# Fabrication of a Modified Polyethersulfone Membrane with Anti-Fouling and Self-Cleaning Properties from SiO_2_-*g*-PHEMA NPs for Application in Oil/Water Separation

**DOI:** 10.3390/polym14112169

**Published:** 2022-05-27

**Authors:** Jun Yin

**Affiliations:** School of Biological and Environmental Engineering, Jingdezhen University, Jingdezhen 333000, China; yinminghan@163.com

**Keywords:** modified polyethersulfone membrane, oil–water emulsion separation, fouling-resistance, self-cleaning, carboxyl-terminated fluorocarbon surfactant

## Abstract

To prepare anti-fouling and self-cleaning membrane material, a physical blending modification combined with surface grafting modification has been carried out; first, poly (2-hydroxyethyl methacrylate) grafted silica nanoparticles (SiO_2_-*g*-PHEMA NPs) were synthesized using surface-initiated activators regenerated by electron transfer atom transfer radical polymerization (ARGET ATRP) and used as a blending modifier to fabricate a polyethersulfone (PES)/SiO_2_-*g*-PHEMA organic–inorganic membrane by the phase-inversion method. During the membrane formation process, hydrophobic PES segments coagulated immediately to form a membrane matrix, and the hydrophilic SiO_2_-*g*-PHEMA NPs migrated spontaneously to the membrane surface in order to reduce interfacial energy, which enhanced the hydrophilicity and anti-fouling properties of the PES/SiO_2_-*g*-PHEMA membrane. Importantly, the membrane surface contained abundant PHEMA segments, which provided active sites for further surface functionalization. Subsequently, the carboxyl-terminated fluorocarbon surfactant (fPEG-COOH) composed of hydrophilic polyethyleneglycol segments and low-surface-energy perfluorinated alkyl segments was synthesized via the esterification of fPEG with succinic anhydride. Lastly, the PES/SiO_2_-*g*-PHEMA/fPEG membrane was prepared by grafting fPEG-COOH onto surface of the PES/SiO_2_-*g*-PHEMA. Thus, a versatile membrane surface with both fouling-resistant and fouling-release properties was acquired. The PES/SiO_2_-*g*-PHEMA/fPEG membrane has a large oil–water flux (239.93 L·m^−2^·h^−1^), almost 21 times that of PES blank membrane and 2.8 times of the PES/SiO_2_-*g*-PHEMA membrane. Compared with the unmodified PES membrane, the flux recovery ratio increased from 45.75% to 90.52%, while the total flux decline ratio decreased drastically from 82.70% to 13.79%, exhibiting outstanding anti-fouling and self-cleaning properties. Moreover, the grafted fPEG segments on the membrane surface show excellent stability due to the presence of stable chemical bonds. The grafted segments remain at the surface of the membrane even after a long shaking treatment. This suggests that this PES/SiO_2_-*g*-PHEMA/fPEG membrane material has potential for application in oil/water separation.

## 1. Introduction

The rapid development of modern industry has resulted in the increasing emission of oily sewage, which may pose risks to the environment. Therefore, environmentally friendly and energy-efficient membrane separation technology has attracted much attention. The polymeric membrane materials were often prepared via the phase-inversion method. Recently, environmentally friendly approaches such as the supercritical phase inversion method have been used in the fabrication of polymeric membrane materials [1].

Among the numerous polymeric membranes that may be used in the treatment of oily sewage, the polyethersulfone (PES) membrane has attracted much attention due to its high resistance to solvents, outstanding hydrolytic stability, excellent resistance to acids and alkalis, and distinguished mechanical strength [2,3,4]. However, the unmodified PES membrane, because of its hydrophobic properties, is vulnerable to membrane fouling, which leads to drastic permeation flux decline during the filtration process.

Membrane surface hydrophilicity has been demonstrated to be the main factor affecting the anti-fouling properties of a fabricated membrane [5,6]. If a hydrophilic membrane surface is present, a dense layer of water molecules can be built on the membrane surface to prevent oil droplets, bacteria, and other pollutants from being adsorbed and deposited on the surface. Thus, the surface exhibits fouling-resistant properties. Therefore, the hydrophilic modification of the membrane surface through surface bioadhesion, surface coating, surface grafting, and physical blending modification has often been attempted [5,6,7,8,9,10,11,12,13,14]. Chitosan was selected as a raw material for the preparation of a hydrophilic chitosan-PEG coating. This material was coated on the polysulfone membrane to provide a membrane with high flux and rejection properties [6]. Regenerated cellulose (RC) has been used as a base membrane material for the fabrication of an RC-*g*-(PNIPAAM-*b*-PPEGMA) membrane with enhanced anti-fouling properties through surface-initiated ATRP [8]. The PES/SiO_2_-*g*-PHEMA membrane has been fabricated via the physical blending of SiO_2_-*g*-PHEMA nanoparticles with PES. Compared with the unmodified PES membrane, the oil/water flux for the modified membrane was seven times greater [12].

However, hydrophilic membrane materials with fouling-resistant properties often exhibit high permeation flux recovery but drastic flux decline during the filtration process [5,6,7,8,9,10,11,12,13,14]. The self-cleaning properties of marine anti-fouling coatings have inspired the development of advanced membrane materials with fouling-release properties through the introduction of low-surface-energy perfluoroalkyl or siloxane segments onto the membrane surface. Low-surface-energy segments at the membrane surface may significantly reduce the interaction between the membrane surface and oil droplets. For the treatment of oily wastewater, oil droplets adsorbed onto the membrane surface can be quickly separated from the membrane surface under the influence of a shearing force. This reduces flux attenuation and endows self-cleaning performance for the membrane [15,16,17,18,19,20,21]. In contrast to the hydrophilic segments, low-surface-energy perfluoroalkane segments do not migrate spontaneously to the surface during the membrane formation process relying on physical blending.

A versatile composite membrane material with both fouling-resistance and fouling-release properties has been prepared. This was achieved through the surface grafting of fluorocarbon surfactant containing hydrophilic polyethylene glycol segments and low-surface-energy perfluoroalkyl segments onto reactive SiO_2_-*g*-PHEMA NPs, which had been prepared by surface-initiated ARGET ATRP.

## 2. Materials and Methods

### 2.1. Materials

PES was provided by JiDa High Performance Materials Co., Ltd. (Jilin, China) and dried before using. Fluorocarbon surfactant (HO(CH_2_CH_2_O)_x_(CF_2_CF_2_)_y_F, fPEG, Mn = 950, 95%) was purchased from DuPont (Shanghai, China). Poly(ethylene glycol) methyl ether (mPEG, M_n_ = 1000, 98%) was purchased from Sigma-Aldrich (Shanghai, China) and used as received. 2-hydroxyethyl methacrylate (HEMA, 98%), bromoisobutyryl bromide (BiBB, 98%), L-ascorbic acid (AsAc, 99%), copper bromide (CuBr_2_, 99%), *N*,*N*,*N*′,*N*″,*N*″-pentamethyldiethylenetriamine (PMDETA, 99%), succinic anhydride (SA, 99%), 3-aminopropyltrimethoxysilane (APTMS, 99%), triethylamine (TEA, 99%), 4-Dimethylaminopyridine (DMAP, 99%), sodium dodecyl sulfate (SDS, 99%), and thionylchloride (SOCl_2_, 99%) were purchased from Aladdin (Shanghai, China) and used without further purification. Dichloromethane (CH_2_Cl_2_), toluene, tetrahydrofuran (THF), dimethylacetamide (DMAc), and 1,4-dioxane were obtained from Aladdin (Shanghai, China) and purified by distillation from sodium hydride.

### 2.2. The Preparation of SiO_2_-g-PHEMA NPs, fPEG-COOH, and PES/SiH1/fPmembrane

The synthetic routes for preparing the SiO_2_-*g*-PHEMA NPs, fPEG-COOH, and PES/SiO_2_-*g*-PHEMA/fPEG membrane are shown in Figure 1. First of all, amino functionalized modified SiO_2_ (SiO_2_-NH_2_) was obtained via the condensation of the hydroxyl groups on the surface of SiO_2_ with trimethoxysilane groups in APTMS. Then, the ATRP initiator-immobilized SiO_2_ NPs (SiO_2_-Br) were prepared via the amidation of SiO_2_-NH_2_ NPs with BIBB. Lastly, the SiO_2_-*g*-PHEMA NPs were prepared by surface-initiated ARGET ATRP. The carboxyl-terminated fluorocarbon surfactant (fPEG-COOH) was synthesized via the esterification of fPEG with succinic anhydride. Lastly, the PES/SiO_2_-*g*-PHEMA/fPEG membrane was prepared by grafting fPEG-COOH onto surface of the PES/SiO_2_-*g*-PHEMA membrane.

#### 2.2.1. Synthesis of SiO_2_-*g*-PHEMA NPs

The SiO_2_ NPs (73 nm) were first prepared according to previous literature [13]. Then, the SiO_2_ NPs were added into toluene and ultrasonic-oscillated to obtain homogeneous suspension. After adding APTMS, the mixture was refluxed under N_2_ for 12 h. Finally, the SiO_2_-NH_2_ NPs were collected by centrifugation and washed with abundant toluene to remove excess APTES. The as-prepared SiO_2_-NH_2_ NPs were dispersed into toluene and ultrasonic-oscillated to obtain homogeneous suspension. Then, a mixture of BIBB and TEA dissolved into toluene was added dropwise into the flask in an ice bath for 2 h. After that, the mixture was agitated at room temperature overnight. Lastly, the SiO_2_-Br NPs were obtained by centrifugation and washed with abundant CH_2_Cl_2_ to remove excess BIBB. In a polymerization flask, the SiO_2_-Br NPs were first dispersed into methanol to produce a homogeneous suspension. After the addition of HEMA, CuBr_2_, and PMDETA, the flask was subjected to three freeze-pump-thaw cycles to remove oxygen. Lastly, AsAc was added to initiate polymerization (the molar ratio of HEMA/CuBr_2_/PMDETA/AA was 5000:1:10:10). The polymerization was proceeded at 50 °C for 12 h; after that, the SiO_2_-*g*-PHEMA NPs were collected and washed with abundant methanol. Lastly, the SiO_2_-*g*-PHEMA NPs were dispersed into DMAc for the next step of the fabrication membrane.

#### 2.2.2. Synthesis of fPEG-COOH and mPEG-COOH

The fPEG-COOH was synthesized through the esterification of fPEG with SA. In a typical procedure, fPEG (16 mmol), SA (40 mmol), DMAP (3.6 mmol), TEA (24.5 mmol), and 1,4-dioxane were added in a flask and stirred. The mixture was warmed to 40 °C and reacted for 24 h under N_2_ protection. After the evaporation of solvent under vacuum, 50 mL saturated NaHCO_3_ aqueous solution was added into the residue. The filtrate was adjusted to pH = 3 after the removal of the precipitate and extracted three times with CH_2_Cl_2_. The organic phase was combined and concentrated to 5 mL, followed by precipitation in 100 mL of cold diethyl ether. The pale-yellow solid was obtained after drying overnight under vacuum at 25 °C (yield, 96%). The mPEG-COOH was prepared via the same steps as above.

#### 2.2.3. Fabrication of PES/SiH1/fPmodified Membrane

In this work, membranes were prepared by the phase-inversion method. The compositions of the casting solution are shown in Table 1. To a dried round-bottomed flask, the SiO_2_-*g*-PHEMA NPs and DMAc were added and ultrasonic-oscillated to obtain homogeneous suspension. Then, the PES was added into the flask and stirred mechanically at 60 °C for 12 h. After the removal of bubble under vacuum, the solution was cast on a clean glass and quickly immersed into a coagulation bath (25 °C, deionized water). The fabricated PES/SiO_2_-*g*-PHEMA membrane was named as PES/SiH1 according to the weight ratio of SiO_2_-*g*-PHEMA to PES. The PES membrane without the addition of NPs was prepared as the blank membrane.

The PES/SiO_2_-*g*-PHEMA/fPEG membrane was prepared through the esterification of the hydroxyl group on the surface of the PES/SiO_2_-*g*-PHEMA membrane with the carboxyl group of fPEG-COOH. In order to improve the surface grafting efficiency, the carboxyl group (-COOH) was firstly transformed into acyl group (-C(O)Cl). In a typical procedure, fPEG-COOH (16 mmol) and SOCl_2_ were added in a flask and stirred, and the solution was refluxed until no gas was generated. After the evaporation of excess SOCl_2_ under vacuum, the residue was dissolved into THF. Then, the as-prepared PES/SiH1 membrane was repeatedly washed with THF and added into a stirred solution of TEA in THF, and fPEG-COCl in THF was added quickly and reacted at 25 °C for 12 h. After being rinsed with abundant THF, the PES/SiO_2_-*g*-PHEMA/fPEG membrane was obtained and named as the PES/SiH1/fP membrane. The PES/SiO_2_-*g*-PHEMA/PEG (PES/SiH1/P) membrane was prepared as the same step.

### 2.3. SiO_2_-g-PHEMA NPs, Polymer and Membrane Characterization

The FTIR spectrums of SiO_2_, SiO_2_-Br, and the SiO_2_-*g*-PHEMA NPs were obtained using the Bruker Tensor 27 spectrometer. The TG analysis of SiO_2_, SiO_2_-Br, and the SiO_2_-*g*-PHEMA NPs was conducted from 50 to 800 °C with under N_2_ using TA Q-600. The top surface morphologies of the fabricated membranes were recorded using Nova NanoSEM 430. The morphology of the SiO_2_-*g*-PHEMA NPs was characterized using Hitachi H-600. ^1^H NMR spectrums were obtained in deuterated chloroform for mPEG, mPEG-COOH, fPEG, and fPEG-COOH using Bruker AV300 MHz. The porosity (ε, %) of the fabricated membrane was obtained according to Equation (1) [2]:(1)$$\varepsilon =\frac{\left(m_{w}-m_{d}\right)}{AL\rho }\times 100\%$$

In Equation (1), *m_w_* (g), *m_d_* (g), *A* (cm^2^), and *L* (cm) were membrane weight (wet), membrane weight (dry), the membrane effective area, and the membrane thickness, respectively. All porosity measurements were carried out at least three times, and the average values were acquired.

The Guerout–Elford–Ferry equation was used to calculate the mean pore size (*r_m_*, nm) of the membrane surface according to Equation (2) [2]:(2)$$r_{m}=\sqrt{\frac{(2.9-1.75\varepsilon )8\eta lQ}{\varepsilon A\Delta P}}$$
where Δ*P* (0.1 MPa), *Q* (m^3^/s), *l* (m), and *η* (Pa·s) were the operational pressure, the volume of permeated pure water per unit time, the membrane thickness, and the pure water viscosity, respectively.

### 2.4. Dynamic and Static Adsorption Test

In order to evaluate the anti-fouling properties of PES, PES/SiH1, PES/SiH1/P, and PES/SiH/fP membranes, each tested membrane that had been rinsed with abundant water was cut into a round shape and fixed into a dead-end filtration system (CB-380); then, the cell was filled with 0.9 g/L oil solution. The dynamic and static adsorption tests were executed under the stirred and unstirred condition, respectively. After incubation 12 h at 25 °C, the amount of adsorbed oil was calculated from the concentrations of oil in the solution before and after adsorption.

### 2.5. Membrane Separation Performance Test

The membrane separation performance test was carried out under a dead-end filtration system (CB-380). Firstly, each membrane was thoroughly rinsed with water and pressurized at 0.2 MPa for 0.5 h to obtain stable pure water flux (PWF). Subsequently, the pressure was set to 0.1 MPa and PWF was measured as *J*_W1_. Then, oil-in-water emulsion (0.9 g·L^−1^, SDS as stabilizer) was conducted via filtration, and the stable oil flux was calculated as *J*_oil_ after 1 h filtration. Finally, the fouled membrane was rinsed thoroughly with water for 0.5 h and PWF was measured as *J*_W2_. *J*_W1_, *J*_W2_ and *J*_oil_ (L·m^−2^·h^−1^) were calculated according to the following equation:(3)$$J=\frac{V}{At}$$
where *V* (L), *A* (cm^2^), and *t* (h) represented the volume of permeated solution, the effective membrane area, and the filtration time, respectively. The oil concentrations of the permeate solution (*C*_p_) and the feed solution (*C*_f_) were obtained by spectrophotometer, and the oil rejection (r, %) was measured as the following equation:(4)$$r=(1-C_{p}/C_{f})\times 100\%$$

The flux recovery ratio (FRR) and total flux decline ratio (*R*_t_) of the fabricated membranes were calculated according to the following equations [3]:(5)$$\mathrm{FRR}\;=\frac{J_{w2}}{J_{w1}}\;\times 100\%$$
(6)$$R_{t}=(\frac{J_{w1}-J_{\mathrm{oil}}}{J_{w1}})\times 100\%$$

## 3. Results and Discussion

### 3.1. The Preparation of SiO_2_-g-PHEMA NPs and fPEG-COOH

Figure 2A displays the FTIR spectra of SiO_2_, SiO_2_-Br, and the SiO_2_-*g*-PHEMA NPs. As for SiO_2_, the peaks at 3440 cm^−1^ and 1100 cm^−1^ are attributed to the O-H and Si-O-Si stretching vibration peaks. After the immobilization of the initiator, a weak absorption peak at 1650 cm^−1^ that belonged to the -NH-C=O group of the initiator appears in the spectrum of the SiO_2_-Br NPs, but this weak peak is not obvious and is easily covered by the broad peak at about 1635 cm^−1^, which is the peak of physically adsorbed water molecules. Moreover, a new absorption peak at 2950 cm^−1^ that belonged to the C-H stretching vibration peak also appears, suggesting the successful immobilization of the initiator. In the spectra of SiO_2_-*g*-PHEMA, the strong peak at 2950 cm^−1^ that was assigned to C-H stretching vibration peak in methyl and methylene groups is clearly visible, and a sharp and strong absorption peak that was assigned to C=O group appears at 1730 cm^−1^. The appearance of these characteristic peaks attributed to PHEMA indicates the successful preparation of the SiO_2_-*g*-PHEMA NPs.

The TGA curves of SiO_2_, SiO_2_-NH_2_, SiO_2_-Br, and the SiO_2_-*g*-PHEMA NPs are shown in Figure 2B. When the SiO_2_ NPs are heated to 800 °C, the mass reduction is about 4.13%, which could be caused by the decomposition of water molecules that physically adsorbed on the surface of SiO_2_ NPs. The mass reduction increases to 12.60% after the immobilization of the initiator onto the surface of the SiO_2_ NPs. In contrast, when the SiO_2_-*g*-PHEMA NPs are heated to 800 °C, the mass reduction is about 65.41%. This could be caused by the thermal decomposition of the PHEMA polymer segments.

Figure 3 displays the TEM images of the unmodified SiO_2_ and the modified SiO_2_-*g*-PHEMA NPs. The unmodified SiO_2_ NPs exhibit severe agglomeration due to their high level of surface energy, while the agglomeration alleviates significantly after grafting PHEMA chains onto the SiO_2_ surface. Apparently, as shown in Figure 3c, the dispersibility of the SiO_2_-*g*-PHEMA NPs improves significantly and large aggregates disappear. Furthermore, the SiO_2_-*g*-PHEMA NPs present a typical core-shell structure, and the exterior surface of the NPs is covered in a uniformly polymer layer with a thickness of about 5 nm (Figure 3d). From the analysis of FT-IR, TGA, and TEM, it is reasonable to believe that the SiO_2_-*g*-PHEMA NPs were prepared successfully.

The surfactant fPEG contains hydrophilic segments (PEG) and low-surface-energy segments (CF_x_), so it was often grafted onto the surface of the material for surface modification [22,23,24]. In this study, fPEG is selected as the membrane-surface modifier. For comparison, mPEG (Mn = 1000), containing only hydrophilic segments, is also grafted onto the membrane surface. The influence of the chain size on the porosity of the membrane surface could be eliminated due to the molar mass of the two molecules that are close to one another. In order to effectively graft fPEG and mPEG molecules onto the surface of the PES/SiH1 composite membrane, mPEG-COOH and fPEG-COOH with carboxyl end group were firstly prepared by esterification reaction. Figure 4A depicts the spectra of fPEG and fPEG-COOH. Compared with fPEG, the new peaks at 2.65 ppm and 4.20 ppm are ascribed to -C(O)-CH_2_CH_2_-C(O)- and -CH_2_-OC=O-, respectively. Moreover, the intensity ratio of the peak **b** and peak **a** is about 2, confirming the successful preparation of fPEG-COOH. Figure 4B shows the spectra of mPEG and mPEG-COOH. In the spectra of mPEG-COOH, the new peaks assigned to -CH_2_-OC=O- (4.20 ppm) and -C(O)-CH_2_CH_2_-C(O)- (2.65 ppm) are observed, which suggests the successful synthesis of mPEG-COOH. In order to improve surface grafting efficiency, the carboxyl group (-COOH) was firstly transformed into the acyl group (-C(O)Cl).

### 3.2. Cross-Section and Surface Morphology of As-Prepared Membranes

The cross-section and surface morphologies of PES, PES/SiH1, and PES/SiH1/fP membranes are shown in Figure 5. Clearly, all tested membranes exhibit asymmetric structures, including a thin dense top-layer, a porous finger-like sub-layer, and fully developed macrovoids at the bottom. The unmodified PES membrane has an obvious macrovoids structure at the bottom. As previously reported, the macrovoids structure frequently appeared in the membranes that had a dense skin layer [2]. As shown in Figure 5d, the unmodified PES membrane has the smallest surface pore size (which will discuss in detail in the porosity section), which obstructs the non-solvent (water molecules) diffusing into the sublayer and results in the formation of macrovoids at the bottom. Compared with the unmodified PES membrane, the morphology of modified PES/SiH1 membrane changes, the macrovoids structure at the bottom is suppressed, and the finger-like pore becomes longer. Since the PES/SiH1 membrane has a larger surface pore size than that of the PES membrane (see Figure 5f), the water molecules diffuse more easily into the sub-layer and induce the formation of many nuclei, resulting in the suppression of the macrovoids.

As shown in Figure 5e, the PES/SiH1 membrane surface contains a number of evenly distributed SiO_2_-*g*-PHEMA NPs due to the spontaneous migration of the hydrophilic SiO_2_-*g*-PHEMA NPs to the surface of the membrane in order to reduce interfacial energy [13]. Thus, the PES/SiH1 membrane surface contains numerous -OH groups, which can provide active sites for further functionalization. After the surface grafting of low-molecular-weight fPEG-COOH (Mn = 1050), the pore size of the PES/SiH1/fP membrane shows no apparent variation with the PES/SiH1 membrane (as shown in Figure 5f,h). Furthermore, the detailed calculation results of the porosity and the mean pore size are discussed in the next sections.

### 3.3. Porosity, Hydrophilicity, and Separation Performance of As-Prepared Membranes

Figure 6 shows the porosity of the PES, PES/SiH1, PES/SiH1/P, and PES/SiH1/fP membranes. Clearly, the PES blank membrane has the smallest porosity (45.23%) and pore size (24.31 nm, according to the Guerout–Elford–Ferry equation). In comparison, the porosity and pore size of the modified PES/SiH1 membrane increases significantly, by approximately 63.12% and 36.25 nm, which could be explained through the following aspects. Firstly, the stability of the casting solution decreases after the addition of SiO_2_-*g*-PHEMA NPs to the casting solution, which results in the formation of the membrane with higher porosity and larger pore size. Secondly, in the membrane formation process, the hydrophilic SiO_2_-*g*-PHEMA NPs migrate spontaneously to the surface of the membrane in order to effectively reduce the interfacial energy, which can enhance the connectivity between the pores and increase the membrane porosity. After grafting the low-molecular-weight (fPEG-COOH, M_w_ = 1050; and mPEG-COOH, M_w_ = 1100) onto the surface of PES/SiH1 membrane, the porosity and pore size of the PES/SiH1/fP membrane (61.72%, 35.52 nm) and the PES/SiH1/P membrane (61.38%, 35.48 nm) decreased slightly, which was in accordance with the surface SEM analysis in Figure 5.

Figure 7A shows the contact angles of the PES, PES/SiH1, PES/SiH1/P, and PES/SiH1/fP membranes. Clearly, the PES blank membrane has the largest water contact angle around 84.5°, suggesting the worst hydrophilicity and the greatest susceptibility to fouling. After the introduction of the SiO_2_-*g*-PHEMA NPs into the casting solution, the surface of the PES/SiH1 membrane enriches an amount of evenly distributed hydrophilic SiO_2_-*g*-PHEMA NPs, resulting in the improvement of the hydrophilicity, and the contact angle reduces to 73.5°. The PES/SiH1/P membrane is prepared via surface grafting hydrophilic polyethylene glycol from the PES/SiH1 membrane. The published literature has proved that the linear polyethylene glycol segments can cooperate with water molecules via the formation of hydrogen bonds, and the crystal form of the water molecules remains unchanged. Thus, the formation of the compact hydration layer on the surface of the PES/SiH1/P membrane by the interaction of the PEG molecules with highly oriented water molecules leads to the smallest contact angle of the PES/SiH1/P membrane (58.2°). By comparison, the contact angle of the PES/SiH1/fP membrane increases slightly (60.3°) because the fPEG molecules contain hydrophobic CF_x_ segments.

The total surface energy of the PES, PES/SiH1, PES/SiH1/P, and PES/SiH1/fP membranes was evaluated by the two-liquid geometric models. As shown in Figure 7B, the unmodified PES membrane has the lowest total surface energy, around 37.6 mJ/m^2^. The total surface energy of the PES/SiH1 membrane increases to 40.8 mJ/m^2^ due to the spontaneous surface segregation behavior of the hydrophilic SiO_2_-*g*-PHEMA NPs (see Figure 5). After the grafting of hydrophilic PEG molecules onto the surface of the PES/SiH1 membrane, the PES/SiH1/P membrane has the highest total surface energy (44.3 mJ/m^2^). Compared with other tested membranes, the total surface energy of the PES/SiH1/fP membrane achieves a minimum value of 32.9 mJ/m^2^ because the low-surface-energy CF_x_ segments were grafted onto the membrane surface.

Figure 8 describes the time-dependent permeate fluxes of the PES, PES/SiH1, PES/SiH1/P, and PES/SiH1/fP membranes. The whole filtration experiment was composed of four steps: 1 h pure water filtration, 1 h oil/water filtration, 0.5 h simple hydraulic washing of the fouled membrane, and 1 h pure water filtration of the washed membrane. Clearly, the unmodified PES membrane has the lowest pure water flux (PWF), only 64.10 L·m^−2^·h^−1^. The PWF of the PES/SiH1 membrane increases to 208.68 L·m^−2^·h^−1^ due to the improvement of the porosity and the hydrophilicity of the PES/SiH1 membrane after the addition of the hydrophilic SiO_2_-*g*-PHEMA NPs as additives. Compared with the PES/SiH1 membrane, the porosity of the PES/SiH1/P membrane decreases slightly due to the surface grafting of the low-molecular-weight mPEG-COOH molecules, while the surface hydrophilicity of the PES/SiH1/P membrane improves significantly. Thus, the PWF of the PES/SiH1/P membrane reaches a maximum (297.23 L·m^−2^·h^−1^). In contrast, the PWF of the PES/SiH1/fP membrane decreases slightly to 278.32 L·m^−2^·h^−1^. In the second part of the oil/water filtration, the permeation fluxes of the PES, PES/SiH1, and PES/SiH1/P membranes declines precipitously within 10 min. Especially, the unmodified PES membrane has the lowest flux (11.09 L·m^−2^·h^−1^) and the highest total flux decline ratio (*R*_t_, 82.7%), while the flux recovery rate (FRR) is only 45.75%, suggesting that the PES membrane is vulnerable to the adsorption of oil droplets and is difficult to alleviate effectively by simple hydraulic cleaning. The oil/water flux of the PES/SiH1 membrane increases to 86.86 L·m^−2^·h^−1^ and the FRR value improves to 69.24%. After grafting the hydrophilic PEG onto the PES/SiH1 membrane, the oil/water flux of the PES/SiH1/P membrane increases to 177.47 L·m^−2^·h^−1^, and the FRR value increases further to 80.21%. Simply increasing the surface hydrophilicity of the membrane (PES/SiH1 and PES/SiH1/P) can effectively induce the adsorption of water molecules to the membrane surface and construct a dense hydration layer between the oil droplets and the membrane surface, blocking the direct contact between the oil droplets and the membrane surface. Thus, the fouling-resistant mechanism is constructed, endowing the anti-fouling properties of the membranes (high FRR value) [12,14]. However, the *R*_t_ values of the hydrophilic membranes are still high (PES/SiH1:58.38%; PES/SiH1/P: 40.29%), which will reduce the separation efficiency and increase the operating costs. In contrast, the *R*_t_ value of the PES/SiH1/fP membrane is the lowest (13.79%) among the tested membranes, and the oil/water flux is the largest (239.93 L·m^−2^·h^−1^), which is almost 21 times that of the PES blank membrane and 2.8 times that of the PES/SiH1 membrane. Compared with the unmodified PES membrane, the value of FRR increases from 45.75% to 90.52%, exhibiting outstanding anti-fouling and self-cleaning properties.

Figure 8B depicts the oil adsorption onto the PES, PES/SiH1, PES/SiH1/P, and PES/SiH1/fP membranes under the unstirred and stirred conditions. Clearly, in the static adsorption test (unstirred), the PES blank membrane has the biggest oil adsorption amount (95.7 µg/cm^2^). Compared with the PES membrane, the oil adsorption amounts of PES/SiH1, PES/SiH1/P, and PES/SiH1/fP membranes decrease obviously. Specifically, the PES/SiH1/P membrane has the smallest oil-adsorption amount (55.2 µg/cm^2^). As discussed in the hydrophilicity section, the linear PEG segments on the surface of the PES/SiH1/P membrane can cooperate with water molecules via the formation of hydrogen bonds and induce the formation of a dense hydration layer to obstruct the adsorption of the oil droplets. In the dynamic adsorption test, the adsorption amounts of all the tested membranes decrease compared to the static adsorption. Interestingly, the PES/SiH1/fP membrane has the smallest adsorption amount, only 32.2 µg/cm^2^ and almost a half of the PES/SiH1/P membrane (50.3 µg/cm^2^). The published literature has proved that the low-surface-energy CF_x_ segments could reduce the interaction with the adsorbed oil droplets and facilitate the removal of oil by the near-surface flow shear under stirring [15].

Figure 9 depicts the anti-fouling and self-cleaning mechanism of the PES/SiH1/fP membrane. The membrane surface is covered with fPEG molecules that contain hydrophilic PEG segments and low-surface-energy CFx segments. The hydrophilic PEG segments can form a hydrogen bond with the water molecules and build a hydration layer on the membrane surface, hindering the adsorption and deposition of the oil droplets and constructing a fouling-resistant mechanism on the membrane surface. At the same time, the low-surface-energy CFx segments on the membrane surface may significantly reduce the interaction between the membrane surface and the oil droplets. For the treatment of oily wastewater, oil droplets adsorbed onto the membrane surface can be quickly separated from the membrane surface under the influence of a shearing force. Thus, a versatile membrane surface with both fouling-resistant and fouling-release properties was acquired.

Table 2 lists the oil/water flux, flux recovery rate, and total flux decline ratio of the PES/SiH1/fP membrane fabricated in this study, and other reported PES-based modified membranes. Compared with other PES-based modified membranes, the PES/SiH1/fP membrane has excellent anti-fouling and self-cleaning properties, which provided theoretical guidance for the development of high-performance advanced membrane materials.

The unmodified PES membrane and the modified membrane (PES/SiH1 and PES/SiH1/fP) are subjected to three complete cycles of filtration tests. As shown in Figure 10A, the initial pure water flux of the PES membrane is 64.10 L·m^−2^·h^−1^, but after three cycles of filtration, it decreases drastically to 17.1 L·m^−2^·h^−1^. After the addition of the hydrophilic SiO_2_-*g*-PHEMA NPs, the PES/SiH1 membrane exhibits the enhanced anti-fouling properties, and the FRR values are 80.21%, 82.89%, and 82.45%, respectively. Unfortunately, the *R*_t_ values are 40.29%, 32.19%, and 30.79%, respectively. The initial pure water flux of the PES/SiH1/fP membrane is 278.32 L·m^−2^·h^−1^, and it is still as high as 209.15 L·m^−2^·h^−1^ after three cycles of filtration. Furthermore, the FRR values are 90.52%, 90.98%, and 91.25%. The *R*_t_ values are 13.79%, 14.02%, and 14.23%, respectively. The results evince that the PES/SiH1/fP membrane has the stable anti-fouling and self-cleaning properties.

In this work, the stability of the PES/SiH1/fP membrane was analyzed through the change of the contact angle of the membrane surface and the dynamic oil/water ultrafiltration experiment. As shown in Figure 10B, the initial contact angle is 60.3° and increases slightly to 61.8° after 30 days of continuous shaking. Furthermore, the PWF and oil/water flux of the membrane before shaking are 278.32 L·m^−2^·h^−1^ and 239.93 L·m^−2^·h^−1^, and the FRR and *R*_t_ values are 90.52% and 13.79%. After shaking for 30 days, these related indexes have almost no change (PWF: 272.19 L·m^−2^·h^−1^, the oil/water flux: 232.34 L·m^−2^·h^−1^, and FRR: 89.54%, *R*_t_: 14.64%), showing that the surface-grafted fPEG segment can stably exist on the membrane surface and provide permanent anti-fouling and self-cleaning properties.

## 4. Conclusions

An amphiphilic porous membrane surface comprised of hydrophilic PEG segments and hydrophobic low-surface-energy CFx segments has been established via physical blending and a surface grafting strategy. The hydrophilic PEG segments can induce the formation of a hydration layer on the membrane surface, hindering the adsorption and deposition of oil droplets, to provide a membrane surface with outstanding anti-fouling properties. The hydrophobic low-surface-energy CFx segments can significantly reduce the interaction between the membrane surface and the oil droplets, so that the attached oil droplets can be quickly separated from the membrane surface under the influence of a shearing force, providing fouling-release action on the membrane surface. In this way, the PES/SiH1/fP membrane displays outstanding anti-fouling and self-cleaning properties. These observations can provide theoretical guidance for the fabrication of high-performance advanced membrane materials.

## Figures and Tables

**Figure 1 polymers-14-02169-f001:**
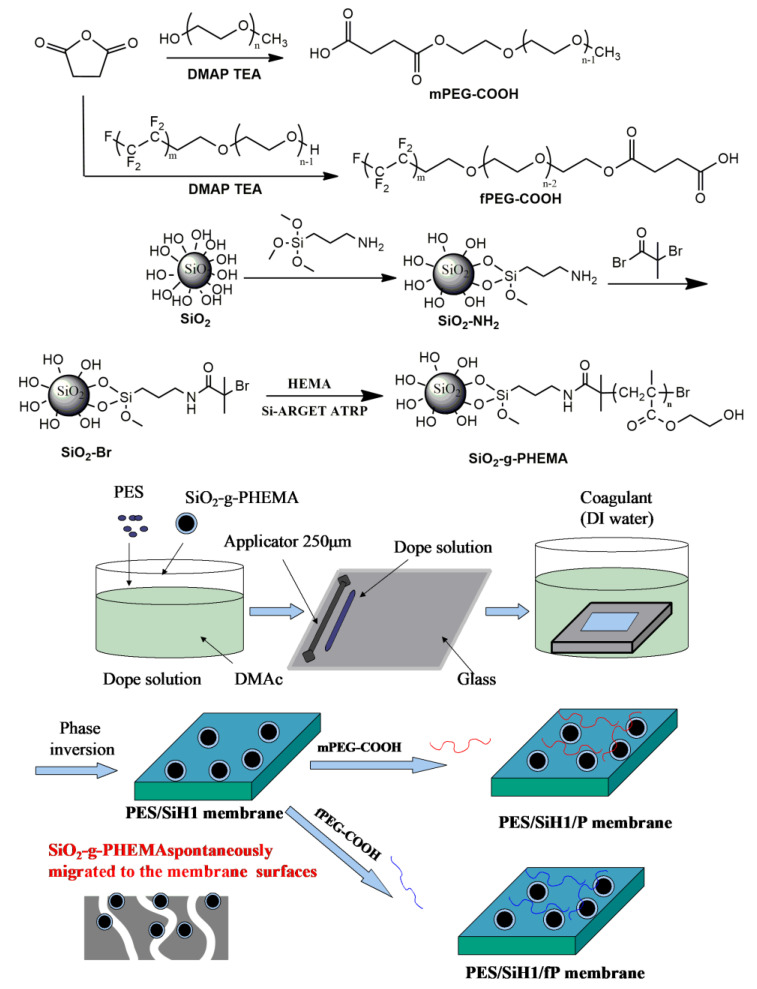
The preparation of the SiO_2_-*g*-PHEMA, fPEG-COOH, and PES/SiH1/fP membrane.

**Figure 2 polymers-14-02169-f002:**
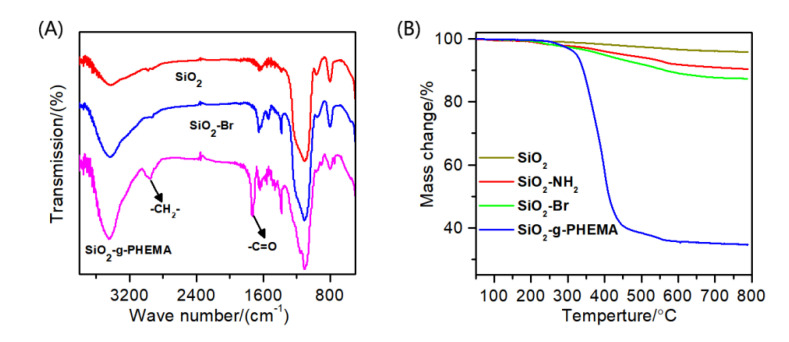
(**A**) The FTIR spectra of SiO_2_, SiO_2_-NH_2_, and the SiO_2_-*g*-PHEMA NPs; (**B**) the TGA curves of SiO_2_, SiO_2_-NH_2_, SiO_2_-Br, and the SiO_2_-*g*-PHEMA NPs.

**Figure 3 polymers-14-02169-f003:**
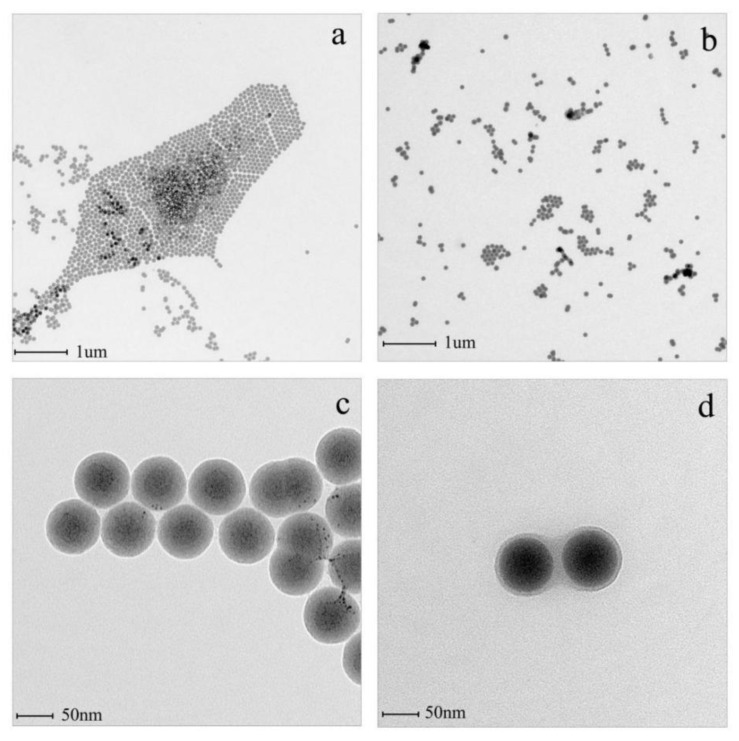
The TEM images of SiO_2_ (**a**,**c**) and the SiO_2_-*g*-PHEMA (**b**,**d**) NPs.

**Figure 4 polymers-14-02169-f004:**
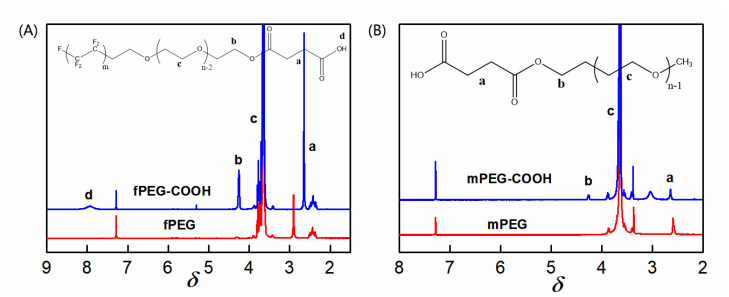
(**A**) The ^1^H-NMR spectra of fPEG and fPEG-COOH; a–d are represents different proton peaks in the chemical structure of fPEG-COOH molecule; (**B**) the ^1^H-NMR spectrum of mPEG and mPEG-COOH; a–c are represents different proton peaks in the chemical structure of mPEG-COOH molecule.

**Figure 5 polymers-14-02169-f005:**
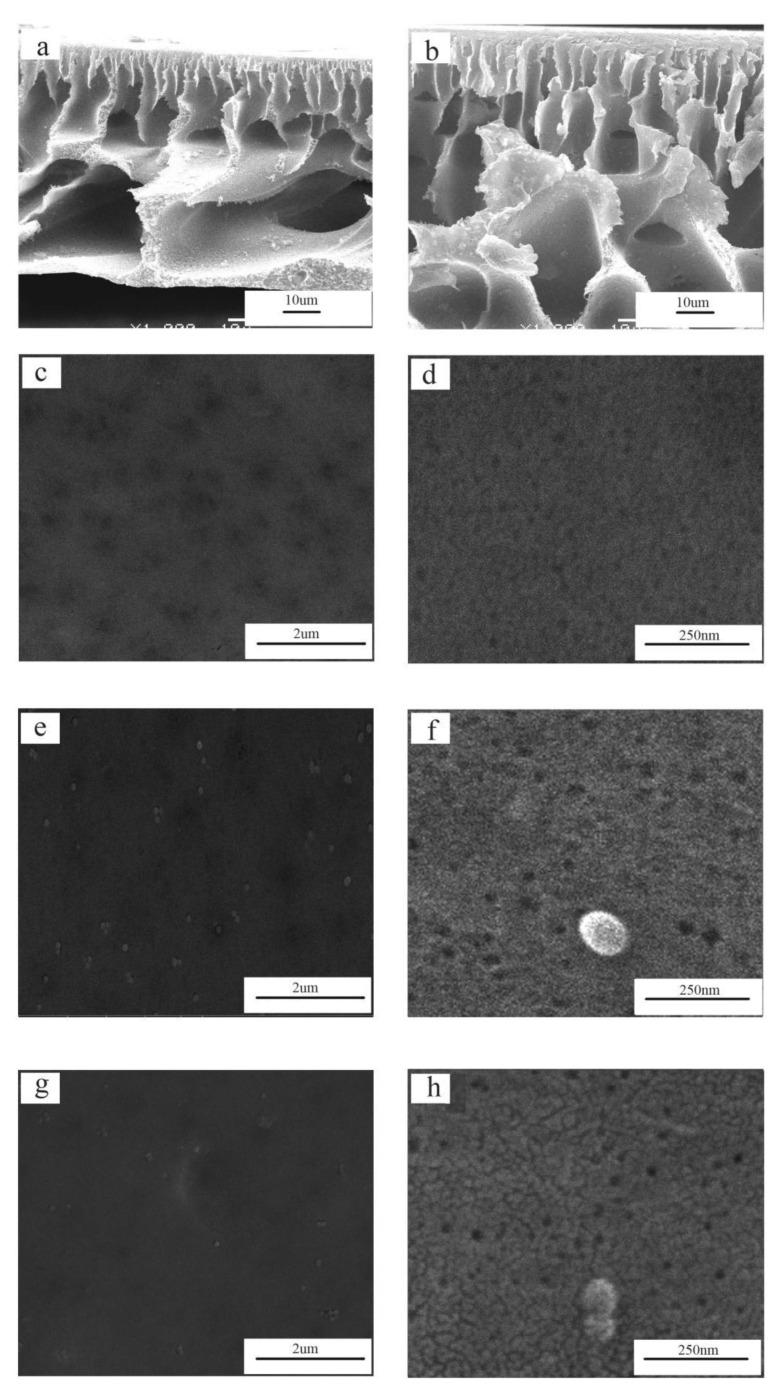
The cross-section morphologies of the PES (**a**) and PES/SiH1 (**b**) membranes; the surface morphologies of the PES (**c**,**d**), PES/SiH1 (**e**,**f**), and PES/SiH1/fP (**g**,**h**) membranes.

**Figure 6 polymers-14-02169-f006:**
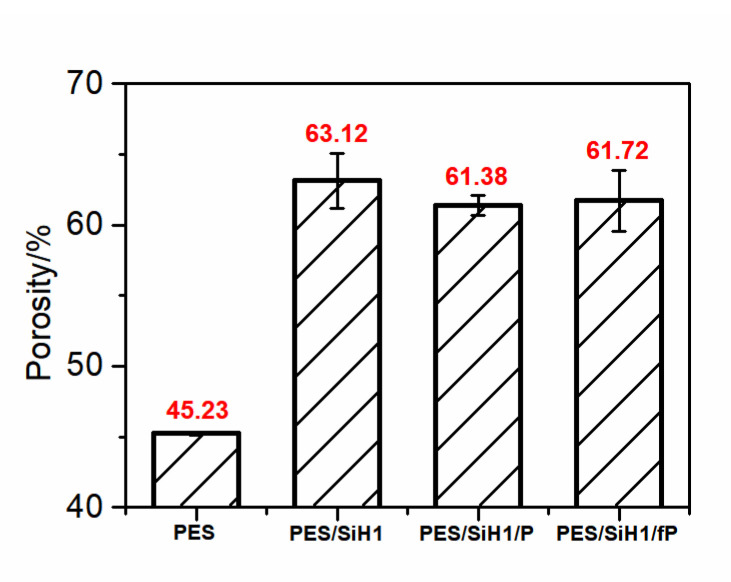
The porosity of the PES, PES/SiH1, PES/SiH1/P, and PES/SiH1/fP membranes.

**Figure 7 polymers-14-02169-f007:**
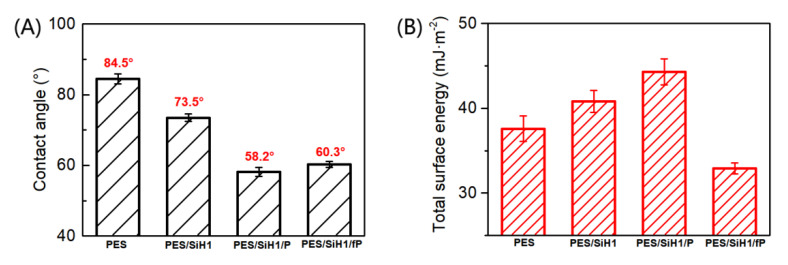
(**A**) The water contact angles of the PES, PES/SiH1, PES/SiH1/P, and PES/SiH1/fP membranes; (**B**) the total surface energy of the PES, PES/SiH1, PES/SiH1/P, and PES/SiH1/fP membrane surfaces calculated by the two-liquid geometric method.

**Figure 8 polymers-14-02169-f008:**
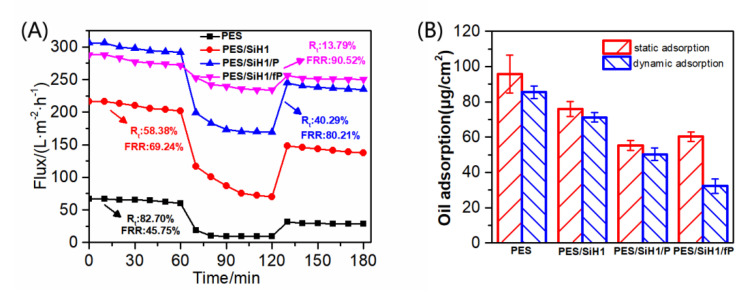
(**A**) The time-dependent fluxes of the tested membranes during the oil-in-water emulsion filtration; (**B**) the oil adsorption onto the PES, PES/SiH1, PES/SiH1/P, and PES/SiH1/fP membranes under unstirred and stirred (200 rpm) conditions.

**Figure 9 polymers-14-02169-f009:**
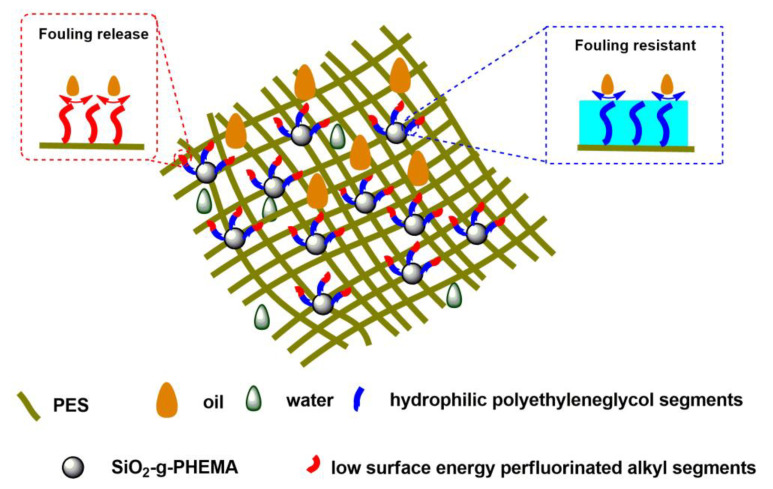
A tentative illustration of the anti-fouling and self-cleaning mechanism of the PES/SiH1/fP membrane.

**Figure 10 polymers-14-02169-f010:**
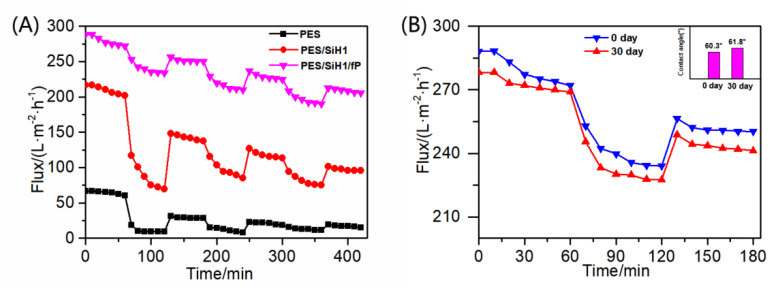
(**A**) The time-dependent fluxes of PES, PES/SiH1, and PES/SiH1/fP membranes during three times of ultrafiltration; (**B**) the time-dependent flux of the PES/SiH1/fP membrane during the oil-in-water emulsion filtration before and after being shaken in water for 30 days.

**Table 1 polymers-14-02169-t001:** The compositions of the casting solution.

Membrane	PES (%)	SiO_2_-*g*-PHEMA (%)	Surface Modifier (Surface Grafting on the PES/SiH1 Membrane)	DMAc (%)
PES	15	/	/	85.00
PES/SiH1	15	0.15	/	84.85
PES/SiH1/P	15	0.15	mPEG-COOH	84.85
PES/SiH1/fP	15	0.15	fPEG-COOH	84.85

**Table 2 polymers-14-02169-t002:** A comparison of the PES/SiH1/fP membrane with other PES membranes on oil–water flux, the flux recovery ratio, and the total flux decline ratio.

Membrane	Oil/Water Flux (L·m^−2^·h^−1^)	FRR (%)	*R*_t_ (%)	Ref.
PES/SiO_2_-*g*-PHEMA	86.86	78.32	55.76	[12]
PES/SiO_2_-*g*-(PDMAEMA-co-PDMAPS)	79.83	84.26	53.67	[14]
PES/Pluronic F127	82.98	63.40	35.00	[25]
F3-PDA/PES	46.10	93.40	20.00	[26]
PES/SiH1/fP	239.93	90.52	13.79	This work

## Data Availability

Not applicable.

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
