# Peer review of "Fabrication of a Modified Polyethersulfone Membrane with Anti-Fouling and Self-Cleaning Properties from SiO2-g-PHEMA NPs for Application in Oil/Water Separation"

_polymers, 2022, doi:10.3390/polym14112169_

Round 1
Reviewer 1 Report
Presented article shows the results of preparation and characterization of polyethersulfone composite membrane with anti-fouling and self-cleaning property from SiO2-g-PHEMA NPs toward oil/water separation. Investigated filler and membranes were characterized by FTIR, SEM, contact angle, TG, 1H NMR, porosity, dynamic and static adsorption test and membrane separation performance test. The presented results are interesting and worth to publish after major revisions and complements.
The remarks and comments are the following:
3.1. The preparation of SiO2-g-PHEMA NPs and fPEG-COOH
Please move the nanoparticle and membrane preparation scheme with description to subsections 2.2-2.4.
3.2. Surface morphology of as-prepared membranes
Please add SEM images of membrane cross sections to show the shape and length of pores inside the membrane. Does an epidermal (non-porous) layer form in the resulting membranes? Does it change after modification on the membrane surface? If so, how does it affect the separation properties of the membrane? Please discuss this topic.
3.3. Porosity, hydrophilicity and separation performance of as-prepared membranes
Please, make pore size measurements by Nitrogen Adsorption Method. Does the pore size change after the investigated modifications? Please provide a discussion on this topic.
Reviewer 2 Report
The manuscript “Fabricating of polyethersulfone composite membrane with anti-fouling and self-cleaning property from SiO2-g-PHEMA NPs toward oil/water separation” deals with the preparation of polymeric membranes via surface-initiated activators regenerated by electron transfer atom transfer radical polymerization and further phase-inversion method. These membranes were characterized by anti-fouling and self-cleaning properties.
The work is well organized and written. Several analyses were performed on the produced membranes, obtaining intriguing results. Therefore, the publication is recommended; but after some revisions, as follows:
- Introduction. The state of the art related to the production of polymeric membranes can be enlarged describing briefly supercritical phase inversion method. For this purpose, see for instance the works of Baldino Lucia et al.; Joseph et al., Green Chemistry Approach for Fabrication of Polymer Composites, https://doi.org/10.3390/suschem2020015; etc…
- Results. Add a brief description before Figure 1. Improve quality of Figure 5. Stress the comparison with the previous literature in the discussion of the present results.
- Conclusions. In the present form, this paragraph is a summary of the results. Rewrite in a more critical way.
- Correct typos.
Reviewer 3 Report
This manuscript reports the modification of poly(ether sulfone) to act as a membrane for oil/water separation. This is an area in which improvement is needed.
The materials prepared are referred to as "composite" membranes. This is unfortunate and may lead to confusion with polymer composites. In this context, "composite" is not necessary and should be avoided.
The manuscript will need significant revision for accuracy, clarity and readability. Corrections are penciled-in directly on pages of the manuscript attached. These are illustrative of the kinds of changes needed throughout. In rewriting, careful attention should be paid to the use of articles, tenses and proper sentence structure. All sentences should be complete. Author's names and personal pronouns should be omitted. "According to previous literature" should be "as previously reported"; round-bottom" should be "round-bottomed"; etc., throughout.

Author Response
Response to Reviewer 3 Comments
Point 1: The materials prepared are referred to as "composite" membranes. This is unfortunate and may lead to confusion with polymer composites. In this context, "composite" is not necessary and should be avoided.
Response 1: Thanks for the reviewer’s comment. We deleted the composite in our revised MS.
Point 2: The manuscript will need significant revision for accuracy, clarity and readability. Corrections are penciled-in directly on pages of the manuscript attached. These are illustrative of the kinds of changes needed throughout. In rewriting, careful attention should be paid to the use of articles, tenses and proper sentence structure. All sentences should be complete. Author's names and personal pronouns should be omitted. "According to previous literature" should be "as previously reported"; round-bottom" should be "round-bottomed"; etc., throughout.
Response 2: Thanks for the reviewer’s comment. We have revised the whole manuscript carefully and tried to avoid any grammar or syntax error. And we have corrected some badly word/constructed sentences and refined the language. Furthermore, we have invited a professor of our university, who is a proficient English speaker, to go through the manuscript and check the English.
Round 2
Reviewer 1 Report
Presented revised article shows the results of preparation and characterization of polyethersulfone composite membrane with anti-fouling and self-cleaning property from SiO2-g-PHEMA NPs toward oil/water separation. The authors rewrote the paper according to the Reviewer suggestion adding the appropriate measurement results and text. I recommend to publish this work.
Author Response
Response to Reviewer 1 Comments
Point 1: Presented revised article shows the results of preparation and characterization of polyethersulfone composite membrane with anti-fouling and self-cleaning property from SiO2-g-PHEMA NPs toward oil/water separation. The authors rewrote the paper according to the Reviewer suggestion adding the appropriate measurement results and text. I recommend to publish this work.
Response 1: Thanks for the reviewer’s comment. We have revised the whole manuscript carefully and tried to avoid any grammar or syntax error.
Reviewer 3 Report
This manuscript is much improved. Some additional minor corrections are needed (see attached).

Author Response
Response to Reviewer 3 Comments
Point 1: This manuscript is much improved. Some additional minor corrections are needed (see attached).
Response 1: Thanks for the reviewer’s comment. We have revised the whole manuscript carefully and tried to avoid any grammar or syntax error.